# High salt diet impairs dermal tissue remodeling in a mouse model of IMQ induced dermatitis

Csenge Pajtók[1], Apor Veres-Székely[1,2], Róbert Agócs[1], Beáta Szebeni[1,2], Péter Dobosy[3], István Németh[4], Zoltán Veréb[4], Lajos Kemény[4], Attila J. Szabó[1,2], Ádám Vannay[2], Tivadar Tulassay[1,2], Domonkos Pap[1,2]*

1 1st Department of Paediatrics, Semmelweis University, Budapest, Hungary, 2 ELKH-SE Pediatrics and Nephrology Research Group, Budapest, Hungary, 3 Institute of Aquatic Ecology, Centre for Ecological Research, Budapest, Hungary, 4 Faculty of Medicine, Department of Dermatology and Allergology, University of Szeged, Szeged, Hungary

* pap.domonkos@med.semmelweis-univ.hu

## Abstract

Recent animal studies, as well as quantitative sodium MRI observations on humans demonstrated that remarkable amounts of sodium can be stored in the skin. It is also known that excess sodium in the tissues leads to inflammation in various organs, but its role in dermal pathophysiology has not been elucidated. Therefore, our aim was to study the effect of dietary salt loading on inflammatory process and related extracellular matrix (ECM) remodeling in the skin. To investigate the effect of high salt consumption on inflammation and ECM production in the skin mice were kept on normal (NSD) or high salt (HSD) diet and then dermatitis was induced with imiquimod (IMQ) treatment. The effect of high salt concentration on dermal fibroblasts (DF) and peripheral blood mononuclear cells (PBMC) was also investigated *in vitro*. The HSD resulted in increased sodium content in the skin of mice. Inflammatory cytokine *Il17* expression was elevated in the skin of HSD mice. Expression of anti-inflammatory *Il10* and *Il13* decreased in the skin of HSD or HSD IMQ mice. The fibroblast marker *Acta2* and ECM component *Fn* and *Col1a1* decreased in HSD IMQ mice. Expression of ECM remodeling related *Pdgfb* and activation phosphorylated (p)-SMAD2/3 was lower in HSD IMQ mice. In PBMCs, production of *IL10*, *IL13* and *PDGFB* was reduced due to high salt loading. In cultured DFs high salt concentration resulted in decreased cell motility and ECM production, as well. Our results demonstrate that high dietary salt intake is associated with increased dermal pro-inflammatory status. Interestingly, although inflammation induces the synthesis of ECM in most organs, the expression of ECM decreased in the inflamed skin of mice on high salt diet. Our data suggest that salt intake may alter the process of skin remodeling.

## Introduction

Salt (NaCl) consumption in modern societies far exceeds the recommendation of the World Health Organization [1].

**Data Availability Statement:** All relevant data are within the manuscript and its Supporting Information files.

**Funding:** T.T. was awarded grant K125470 from the National Research, Development and Innovation Office (NKFI) (https://nkfih.gov.hu/). A. J.Sz. was awarded grant 2020-4.1.1-TKP2020 from the Ministry for Innovation and Technology (https://kormany.hu/innovacios-es-technologiai-miniszterium); grant STIA-KFI-2020 from Semmelweis Science and Innovation Fund (https://semmelweis.hu) and grant 20382-3/2018 FEKUTSTRAT from NKFI. D.P. was awarded grant NKFI 124549. The funders had no role in study design, data collection and analysis, decision to publish, or preparation of the manuscript.

**Competing interests:** The authors have declared that no competing interests exist.

**Abbreviations:** ACTA2, α-SMA: alpha smooth muscle actin; ACTB, beta-actin; bp, base pair; COL1A1, collagen-1a1; DF, dermal fibroblast; ECM, extracellular matrix; F, forward; FN, fibronectin; GAPDH, glyceraldehyde 3-phosphate dehydrogenase; HSD, high salt diet; IL, interleukin; IMQ, imiquimod; MRI, magnetic resonance imaging; NSD, normal salt diet; PBMC, peripheral blood mononuclear cell; PCR, polymerase chain reaction; PDGF-B, platelet-derived growth factor; R, reverse; Ta, annealing temperature; TGF-ß, transforming growth factor beta; TNF-α, tumor necrosis factor alpha; VCL, vinculin; VIM, vimentin.

Sodium ($Na^+$) homeostasis is classically considered to be regulated by the kidneys. However, several previous human and animal studies demonstrated that sodium is stored in the skin in both free and inactive forms [2–9]. In addition, our previous study demonstrated that dermal fibroblasts are important mediators of skin sodium storage [10].

Beside the well-known role of sodium in maintaining body fluid homeostasis and blood volume recent studies demonstrated that excessive salt intake can trigger or worsen inflammatory mechanisms in various organs, as well [11]. Indeed, it has been shown that a high-salt diet enhances inflammation in the lung, kidney, gut, central nervous system and promotes progression of autoimmune diseases [4,12–16]. One of the best characterized salt-related pro-inflammatory effects is the increased differentiation of naive T cells toward T helper 17 (Th17) cells and their elevated IL-17 production [17,18].

Increasing number of experimental data suggest that local sodium excess mediates pathological processes in the skin. In fact, regions affected by atopic dermatitis have been shown to contain more sodium than healthy regions [19]. Furthermore, there is a correlation between increased dietary salt intake and aggravation of symptoms of psoriasis in humans [17]. It has also been reported that increased salt intake can delay wound closure in the skin [20]. Taken together these data indicate that increased salt intake has a more complex effect on the pathological processes in the skin than previously thought, however little is known about the underlying mechanisms.

Our aim was to reveal how salt consumption affects the skin pathology, including inflammation and related tissue remodeling process. Therefore, we examined the effect of increased salt intake on the skin expression of inflammatory cytokines and markers of tissue remodeling using the mouse model of imiquimod (IMQ) -induced dermatitis. We also examined the effect of salt loading on the functional activity and molecular biological changes of primary dermal fibroblasts, which play a central role in the extracellular matrix (ECM) production in the skin.

## Methods

### Animal experiments

All experiments were approved by the Committee on the Care and Use of Laboratory Animals of the Council on Animal Care at the Semmelweis University of Budapest, Hungary (PEI/001/1731-9/2015). Experiments were performed on male C57BL/6J mice which were 7–8 weeks of age. Animals were housed in a temperature-controlled (22±1˚C) room with alternating light and dark cycles (12/12 hours). Mice were randomly divided into four groups: normal salt diet (NSD) group; normal salt diet and imiquimod treatment (NSD IMQ) group; high salt diet (HSD) group; high salt diet and imiquimod treatment (HSD IMQ) group (n = 6-8/groups) (Fig 1A). Two types of dietary regimes were applied for 5 weeks: normal-salt diet (NSD) consisting of standard rodent chow and high-salt diet (HSD) consisting of $Na^+$ rich rodent chow (8.0% NaCl; Ssniff Gmbh, Soest, Germany). Mice had *ad libitum* access to food and water. On the fifth week of the experiment mice in groups NSD IMQ and HSD IMQ received daily topical 62.5 mg Aldara cream (5% imiquimod; Meda AB, Solna, Sweden) on their depilated back for 5 consecutive days. The area of the topical IMQ-installation was 6 $cm^2$. Control mice of NSD and HSD groups were treated similarly with vaseline. The general condition of the mice was monitored daily by visual examination during the entire experiment. After the termination of the experiment, skin was displaced from the back and snap-frozen in liquid nitrogen for further molecular biological measurements. All surgical procedures were performed under total anesthesia by the intraperitoneal injection of a mixture of 100 mg/kg ketamine (Richter Gedeon, Budapest, Hungary) and 10 mg/kg xylazine (Medicus Partner, Biatorbágy, Hungary). The mice were euthanized by overdose of anesthetic solution.

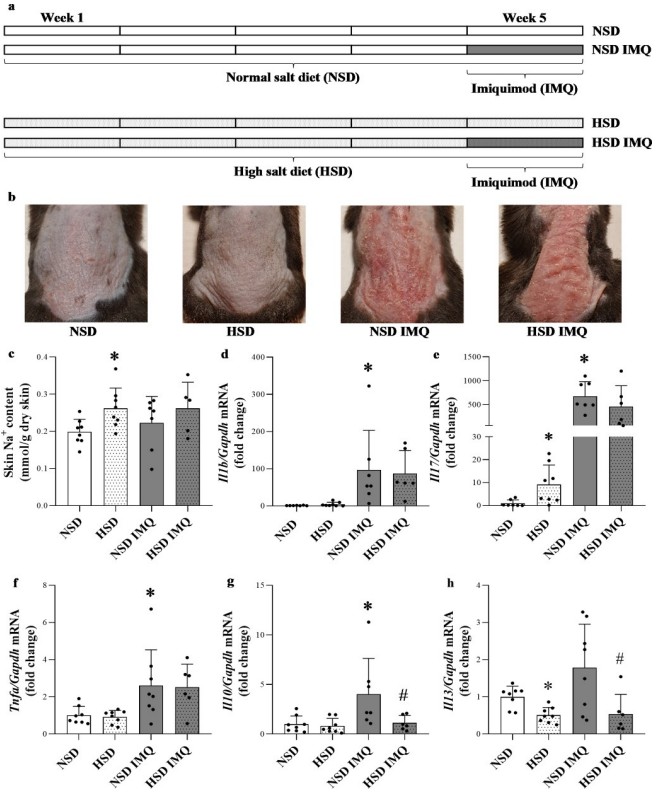

**Fig 1. Effect of high salt diet on dermal inflammation in IMQ-treated and control mice.** (**a**) Mice were kept on NSD or HSD for 5 weeks and on the last 5 days of the dietary regime the mouse model of IMQ-induced dermatitis was performed on the indicated group of animals. (**b**) 5 days of IMQ (62.5mg with 5% imiquimod cream) or vaseline treatment applied to NSD and HSD mice resulted in the inflammation of the back skin of mice. (**c**) The skin $Na^+$ content was measured by flame photometry following tissue digestion. (**d-h**) The dermal mRNA expression of *Il1b*, *Il17*, *Tnfa*, *Il10* and *Il13* in the skin tissues of mice was determined by real-time PCR in comparison with *Gapdh*. Data were normalized and presented as the ratio of the mean values of the control group. Values were expressed as mean +SD. Dots represent individual values. n = 6–8 in each group; *p<0.05 vs. NSD; #p<0.05 vs. NSD IMQ; (Unpaired t-test (c,f,h), Mann-Whitney U-test (d,e,g)).

## Determination of skin $Na^+$ content

Skin samples were desiccated at 90˚C for 24 hours. Skin samples were digested in a mixture of 5 cm³ Suprapur® 65% nitric acid (VWR International, Pennsylvania, USA) and 2 cm³ Suprapur® 30% hydrogen-peroxide (VWR International) applying a TopWave microwave assisted digestion system (Analytik Jena, Jena, Germany). After this procedure solution volume was made up to 25 cm³ with de-ionized water, then $Na^+$ content was determined by using a FP910 flame photometer (PG Instruments, Lutterworth, UK).

## Human primary dermal fibroblasts (DF)

Human dermal fibroblasts were obtained from healthy individuals after signing an informed consent. Collection of biospecimens for the isolation and culture on human dermal fibroblasts complied with the directive of the Helsinki Declaration and were approved by the local Ethics Committee of the University of Szeged, Hungary (CSR/039/00346-5/2015). Tissue samples were collected corresponding to the EU Member States' Directive 2004/23/EC on tissue isolation as well. Primary human DFs were isolated from a healthy individual undergoing plastic surgery [21]. Fibroblast cells were cultured in Dulbecco's modified Eagle's medium (DMEM;

Gibco, Thermo Fisher Scientific, Waltham, Massachusetts, USA) supplemented with a 10% fetal bovine serum (FBS; Gibco, Thermo Fisher Scientific), 1% L-glutamine and 1% penicillin/ streptomycin, incubated in a humidified incubator with 5% $CO_2$ at 37˚C. Before the *in vitro* experiments, the medium was replaced with DMEM without FBS for 24 hours. For real-time PCR measurements DF cells were seeded into 96-well plates (Sarstedt, Nümbrecht, Germany) at a density of $10^4$ cells/well (n = 6 well/treatment group) and were subjected to culture medium with normal (Control; 150 mmol/L) or high sodium concentration (NaCl; 200 mmol/ L) with or without recombinant human TGF-ß (1 nmol/L) (Thermo Fisher Scientific). The additional sodium concentration was determined as previously described [10]. Vehicle treated cells served as controls. After 24 hours the cells were collected for real time PCR measurements.

## Peripheral blood mononuclear cells (PBMCs)

Isolation and culture on human PBMCs were approved by Semmelweis University Regional and Institutional Committee of Science and Research Ethics (31224-5/2017/EKU) after signing an informed consent. PBMCs from healthy adult donor were isolated by density gradient centrifugation using Histopaque-1077 (Merck KGaA, Darmstadt, Germany). After isolation, the cells were placed into RPMI 1640 medium (ATCC, Manassas, Virginia, USA) supplemented with 10% FBS (Gibco, Thermo Fisher Scientific) and 1% penicillin/streptomycin solution in humidified 95% air and 5% $CO_2$ at 37˚C. Before the *in vitro* experiments, the medium was replaced with RPMI 1640 medium without FBS for 24 hours. For real time PCRs, PBMCs were seeded into 24 well plates (Sarstedt) at a density of $5 \times 10^5$ cells/well (n = 5 well/treatment group) and were subjected to culture medium with normal (Control; 150 mmol/L) or high sodium concentration (NaCl; 200 mmol/L).

## RNA isolation and cDNA synthesis

Total RNA was isolated from frozen skin samples, PBMCs and DFs by Total RNA Mini Kit (Geneaid Biotech Ltd., New Taipei City, Taiwan) according to manufacturer's instructions. The concentration and quality of the isolated RNA were determined by DeNovix DS-11 spectrophotometer (DeNovix Inc., Wilmington, Delaware, USA). 2,500 ng of total RNA from skin and 50 ng of total RNA from PBMCs and DFs were reverse-transcribed using Maxima First Strand cDNA Synthesis Kit for real time PCR (Thermo Fisher Scientific) to generate firststranded cDNA.

## Real-time polymerase chain reaction (PCR)

Real-time PCRs were performed in a final volume of 20 μl containing 0.5 μM of forward and reverse primers (Integrated DNA Technologies, Coralville, Iowa, USA), 10 μl of Light Cycler 480 SYBR Green I Master enzyme mix (Roche Diagnostics, Mannheim, Germany) and 1 μl cDNA on a LightCycler 480 system (Roche Diagnostics). The nucleotide sequences of the applied primer pairs were designed as previously described [22]. Their specific optimal annealing temperatures and product lengths are summarized in Table 1. Results were analyzed by Light-Cycler 480 software version 1.5.0.39 (Roche Diagnostics). The mRNA expressions were determined by comparison with the expression of *GAPDH* or *RPLP0* as a housekeeping gene from the same samples. The data were normalized and presented as the ratio of the mean values of their control groups.

**Table 1. Nucleotide sequences of primer pairs, product length and specific annealing temperatures (Ta) applied for the real-time reverse transcriptase polymerase chain reaction (RT- PCR) detection.**

| Gene | | Primer pairs | Product lenght | T$_a$ |
|---|---|---|---|---|
| *mouse GAPDH* | F: | 5'- ATC TGA CGT GCC GCC TGG AGA AAC -3' | 164 bp | 60˚C |
| | R: | 5'- CCC GGC ATC GAA GGT GGA AGA GT -3' | | |
| *mouse ILβ* | F: | 5'- GCC ACC TTT TGA CAG TGA TGA GAA -3' | 136 bp | 55˚C |
| | R: | 5'- GAT GTG CTG CTG CGA GAT TTG A -3' | | |
| *mouse IL10* | F: | 5'- CAA AGG ACC AGC TGG ACA ACA TAC -3' | 124 bp | 55˚C |
| | R: | 5'- GCC TGG GGC ATC ACT TCT ACC -3' | | |
| *mouse IL17* | F: | 5'- AGG ACT TCC TCC AGA ATG T -3' | 136 bp | 60˚C |
| | R: | 5'- TCA GGG TCT TCA TTG CGG -3' | | |
| *mouse TNFα* | F: | 5'- GGG CCA CCA CGC TCT TCT GTC TA -3' | 83 bp | 56˚C |
| | R: | 5'- GCG CGG GAG GCC ATT TGG GAA CTT -3' | | |
| *mouse ACTA2* | F: | 5'- CCC CTG AAG AGC ATC GGA CA -3' | 105 bp | 60˚C |
| | R: | 5'- TGG CGG GGA CAT TGA AGG T -3' | | |
| *mouse FN* | F: | 5'- GGT CAG GGC CGG GGC AGA T -3' | 228 bp | 60˚C |
| | R: | 5'- CTG GCT GGG GGT CTC CGT GAT AAT -3' | | |
| *mouse COL1A1* | F: | 5'- CAA AGG TGC TGA TGG TTC T -3' | 107 bp | 60˚C |
| | R: | 5'- GAC CAG CTT CAC CCT TG -3' | | |
| *mouse PDGFB* | F: | 5'- CTG GGC GCT CTT CCT TCC TCT C -3' | 170 bp | 60˚C |
| | R: | 5'- CCA GCT CAG CCC CAT CTT CAT C -3' | | |
| *mouse IL13* | F: | 5'- ACG GCA GCA TGG TAT GGA GTG TGG -3' | 161 bp | 55˚C |
| | R: | 5'- GGG GAG GCT GGA GAC CGT AGT GG -3' | | |
| *mouse TGFβ* | F: | 5'- GTG CGG CAG CTG TAC ATT GAC TTT -3' | 239 bp | 60˚C |
| | R: | 5'- GGC TTG CGA CCC ACG TAG TAG AC -3' | | |
| *human GAPDH* | F: | 5'- AGC AAT GCC TCC TGC ACC ACC AA -3' | 159 bp | 60˚C |
| | R: | 5'- GCG GCC ATC ACG CCA CAG TTT -3' | | |
| *human IL10* | F: | 5'- ATG CCC CAA GCT GAG AAC CAA GAC -3' | 107 bp | 60˚C |
| | R: | 5'- AGA AAT CGA TGA CAG CGC CGT AGC -3' | | |
| *human IL13* | F: | 5'- CTG CAA ATA ATG ATG CTT TCG A -3' | 90 bp | 54˚C |
| | R: | 5'- CCA GTT TGT AAA GGA CCT GCT CT -3' | | |
| *human FN* | F: | 5'- GGC TGC CCA CGA GGA AAT CTG C -3' | 229 bp | 58˚C |
| | R: | 5'- GTG CCC CTC TTC ATG ACG CTT GTG -3' | | |
| *human COL1A1* | F: | 5'- CTG CCC CGG CGC CGA AGT C -3' | 96 bp | 60˚C |
| | R: | 5'- CCC TCG ACG CCG GTG GTT TCT TG -3' | | |
| *human VIM* | F: | 5'- GAG GCT GCC AAC CGG AAC AAT GAC -3' | 203 bp | 59˚C |
| | R: | 5'- TCC TGC AGG CGG CCA ATA GTG TCT -3' | | |
| *human VCL* | F: | 5'- CCA CGG CGC CTC CTG ATG C -3' | 152 bp | 60˚C |
| | R: | 5'- GGC CTG AAT GCC TTC CAC TGT TGA -3' | | |
| *human ACTB [22]* | F: | 5'- ACC GAG CGT GGC TAC AGC TTC ACC -3' | 114 bp | 53˚C |
| | R: | 5'- AGC ACC CGT GGC CAT CTC TTT CTC G-3' | | |

## Protein isolation and Western blot analysis

Dermal tissue samples from mice were homogenized in lysing solution, containing 50 mM HEPES, 150 mM NaCl, 1% Triton X-100, 5 mM EDTA, 5 mM EGTA, 20 mM sodium pyrophosphate, 20 mM NaF, 0.2 mg/mL phenylmethylsulfonyl fluoride, 0.01 mg/mL leupeptin, and 0.01 mg/mL aprotinin (pH 7.4) (each substance obtained from Merck). Thereafter protein concentration was determined by a detergent-compatible protein assay (BioRad, Hercules, CA, USA). Denatured samples (20 µg protein/lane) were loaded and separated on 4–20%

gradient SDS polyacrylamide gel (BioRad) and transferred to nitrocellulose membranes (BioRad). Pre-stained protein standard (BioRad) was used as molecular weight marker. To verify the transfer, membranes were stained with Ponceau S (Sigma-Aldrich), then washed and blocked with 5% non-fat milk in TRIS-buffered saline (TBS) for 1 h at RT. Thereafter, they were incubated overnight at 4°C with antibodies specific for phosphorylated (p)-SMAD2/ 3 (sc-11769; 1:500, Santa-Cruz Biotechnology) or GAPDH (sc-47724; 1:2000, Santa Cruz Biotechnology). After repeated washing with TBS containing 0.05% Tween-20 and 1% non-fat milk, membranes were incubated with the corresponding horseradish peroxidase (HRP)- conjugated secondary antibodies (1:1000 anti-rabbit or 1:5000 anti-mouse, Santa Cruz Biotechnology) for 1 h at RT. Immunoreactive bands were developed using enhanced chemiluminescence detection (Immobilon Forte Western HRP substrate, Merck KGaA) and the resulted images were visualized with VersaDoc 5000MP system (Bio-Rad) and analyzed using Quantity One v4.6.9 (Bio-Rad) and ImageJ 1.48v (National Institutes of Health, Bethesda, MD, USA) softwares. Relative protein levels were determined by comparison with GAPDH as protein loading control. Data were normalized and presented as the ratio of the mean values of the control group.

### SiriusRed collagen detection assay

For the SiriusRed assay, human DF cells were seeded into 96-well plates (Sarstedt) at a density of $10^4$ cells/well (n = 5 well/ treatment group) and were subjected to culture medium with normal (Control; 150 mmol/L) or high sodium concentration (NaCl; 200 mmol/L) with or without recombinant human TGF-ß (1 nmol/L) (Thermo Fisher Scientific) for 24 hours in 37°C. The assay was performed as previously described [23]. All reagents were purchased from Merck KGaA. Absorbance was determined at 544 nm by Hidex Chameleon Microplate Reader (Lablogic Systems, Sheffield, United Kingdom) using the MikroWin 2000 software.

### Immunocytochemistry and cell morphology analysis

Human DF cells were seeded in cell culture chambers (Sarstedt) at a density of $10^4$ cells/well and were subjected to culture medium with normal (Control; 150 mmol/L) or high sodium concentration (NaCl; 200 mmol/L) for 24 hours in 37°C. After washing with WashPerm solution, slides were permeabilized with Cytofix/Cytoperm (BD Pharmingen, San Diego, California, USA) at room temperature for 15 minutes. Slides were incubated with primary antibody specific for α-SMA (mouse, 1:1000; MERCK, Darmstadt, Germany) at room temperature for 1 hour. Thereafter the slides were washed and incubated with corresponding Alexa Fluor 488 conjugated secondary antibody (anti-mouse, 1:100; Invitrogen, Thermo Fisher Scientific) at room temperature in the dark for 30 minutes. Finally, slides were coverslipped with ProLong Gold antifade reagent (Invitrogen, Thermo Fisher Scientific). Appropriate controls were performed by omitting the primary antibodies to assure their specificity and to avoid autofluorescence. Assessment of the circularity of α-SMA immunopositive DFs was carried out by graphical analysis using ImageJ 1.48v software. Images of 35–40 randomly selected cells from each treatment group were taken using the Olympus IX81 fluorescent microscope (Olympus, Japan). The circularity of the fibroblasts was calculated based on the following formula: circularity = $4^*\pi$(area/perimeter^2).

### Migration assay

Human dermal fibroblasts were seeded at a density of $2x10^4$ cells/well (n = 8 well/ treatment group) into 96 well-plates (Sarstedt) containing non-toxic gel barriers to create cell-free zones. After 24 hours of incubation, barriers were removed and wells were washed with PBS, thereafter cells were treated with culture medium containing normal (Control; 150 mM) or high sodium

concentration (NaCl; 200 mM). Bright-field images of each well were taken using Olympus IX81 microscope system (Olympus) at 0, 24 and 48 hours after treatment. Cell-free gap area was measured using ImageJ 1.48v software and determined as a ratio of initial gap area at 0 hour.

## Statistical analysis

Statistical evaluation of data was performed by GraphPad Prism 7.0 Software (GraphPad Software, La Jolla, California, USA). To test if the the groups show a normal distribution, Kolmogorov-Smirnov normality test was performed. Data were analyzed by unpaired t-test for normal distribution or by Mann-Whitney U-test for non-normal distribution. Multiple comparisons of the raw data derived from migration assays were performed by using multiple t-test and ordinary two-way ANOVA with Dunnett correction. Data were normalized and presented as the ratio of the mean values of their control groups. $p \leq 0.05$ was considered as statistically significant. Values were expressed as mean+SD.

## Results

### Effect of high salt diet on dermal inflammatory cytokine expression

We examined the effect of 8% NaCl diet on the body weight of mice and found no significant difference in body weight between animals on NSD and HSD (S1 Fig), which is consistent with the results of other research groups [24–27]. In order to examine the effect of high salt diet on dermal inflammation imiquimod (IMQ) treatment was performed on mice receiving normal (NSD) or high salt diet (HSD) (Fig 1A). In accordance with previous data HSD increased the Na$^+$ content of the skin (Fig 1C) [5,8]. Furthermore, IMQ application induced remarkable dermatitis, as expected (Fig 1B). Indeed, IMQ increased the dermal mRNA expression of *Il1b*, *Il17*, *Tnfa* and *Il10* (Fig 1D–1G) in NSD group. The synthesis of pro-inflammatory *Il17* and anti-inflammatory *Il10* and *Il13* was changed in the skin by high salt intake. Indeed, dermal mRNA expression of *Il17* was increased in HSD group compared to NSD (Fig 1E). The mRNA expression of *Il10* was lower in the skin of HSD IMQ group compared to that of NSD IMQ group (Fig 1G). Furthermore, mRNA expression of *Il13* was decreased in both HSD and HSD IMQ group compared to that of NSD and NSD IMQ group (Fig 1H).

### Effect of high NaCl environment on the anti-inflammatory and profibrotic cytokine production of human PBMCs

To investigate whether the high salt loading is responsible for the decreased production of immune cell- derived anti-inflammatory *IL10*, *IL13* and profibrotic *PDGF-B* in the skin, we investigated its effect on PBMCs. We found, that high NaCl loading inhibited the mRNA expression of *IL10*, *IL13* and *PDGF-B* of human PBMCs (Fig 2A and 2B).

### Effect of high salt diet on dermal ECM remodeling

To describe the effect of excess dietary salt intake on tissue remodeling ECM markers were studied in the above experimental setting (Fig 1A). We found that dermal mRNA expression of *Acta2*, *Col1a1* and *Fn* was decreased in HSD IMQ group compared to that of NSD IMQ group (Fig 3A–3C). In addition to the ECM markers, ECM remodeling-related growth factors and enzymes in vehicle and IMQ-treated mice were also investigated. We found that dermal mRNA expression of *Pdgfb* was decreased in HSD and HSD IMQ groups compared to that of NSD and NSD IMQ groups. (Fig 3D). *Tgfb* was not affected by salt diet, but the IMQ treatment increased its amount in the skin (Fig 3E). Moreover, the protein amount of phosphorylated (p)SMAD2/3 was significantly decreased in HSD IMQ group compared to NSD IMQ group (Fig 3F).

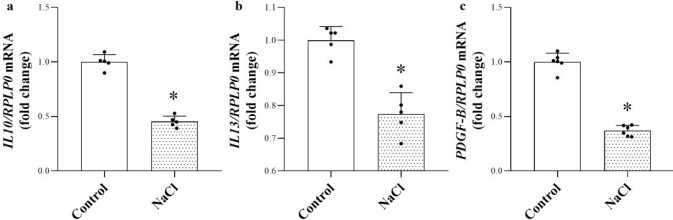

**Fig 2. Effect of high NaCl environment on the anti-inflammatory and profibrotic cytokine production of human PBMC. (a-b)** The mRNA expression of anti-inflammatory *IL10*, *IL13* and profibrotic *PDGF-B* were measured by real-time PCR in comparison with *RPLP0*. Data were normalized and presented as the ratio of the mean values of the control group. Values were expressed as mean+SD. Dots represent individual values. n = 5 in each group; *p<0.05 vs. Control (Unpaired t-test).

## Effect of high NaCl environment on the cell motility of human primary dermal fibroblasts

The effect of increased NaCl concentration on cellular morphology was visualized by α-SMA immunofluorescence staining on the cytoskeleton of human DFs *in vitro*. While control cells had elongated shape, on high salt loaded cells flattened and more circular morphology was

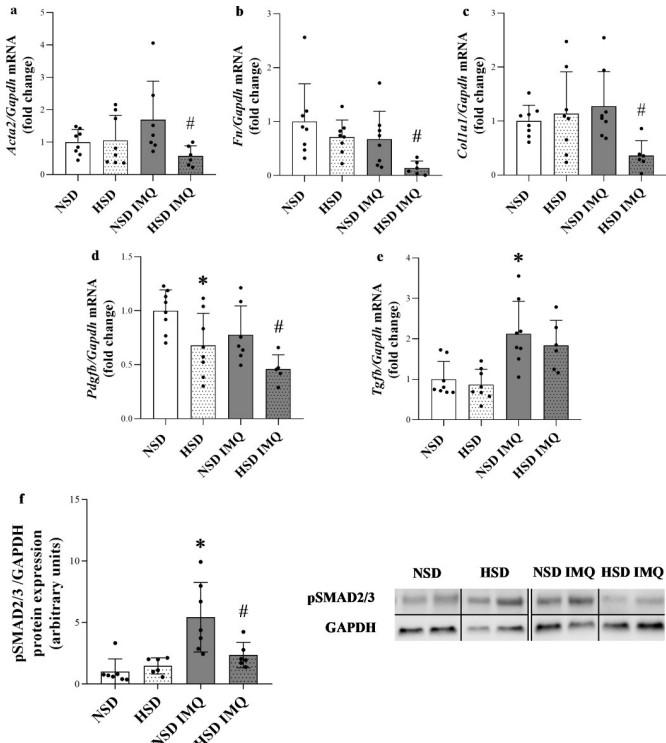

**Fig 3. Effect of high salt diet on ECM marker production and dermal tissue remodeling-associated factors of IMQ-treated and control mice. (a-e)** The dermal mRNA expression of *Acta2*, *Fn*, *Col1a1*, *Pdgfb* and *Tgfb* in the skin tissues of mice was determined by real-time PCR in comparison with *Gapdh*. (**f**) The dermal protein expression of (p) SMAD2/3 in the skin tissues of mice was determined by Western blot in comparison with GAPDH as protein loading control. Protein samples from control and IMQ-treated mice were run on separate gels. They were compared based on internal control. Full-length blots are included in *Supporting Information*. Data were normalized and presented as the ratio of the mean values of the control group. Values were expressed as mean+SD. Dots represent individual values. *n* = 6–8 in each group; *p<0.05 vs. NSD; #p<0.05 vs. NSD IMQ; (Unpaired t-test (a,b,d), Mann-Whitney U-test (c,e,f)).

observed. Accordingly, quantitative determination of the shape of dermal fibroblast cells showed significantly elevated circularity in the high salt treated group (Fig 4A). Furthermore, the mRNA expression of fibroblasts motility regulator *VIM*, *VCL* and *ACTB* decreased in the high salt treated group (Fig 4B–4D). Consistent with these results, larger cell-free area indicating reduced migration capacity was observed in the high salt-treated DFs at both 24 and 48 hours of the experiment (Fig 4E).

## Effect of high NaCl environment on the ECM marker production of human primary dermal fibroblasts

Since DFs are the main sources of skin ECM, we examined the direct effect of salt loading on their functional activity. Consistent with our *in vivo* results in IMQ-treated animals, increased salt concentration in culture media reduced both endogenous and TGF-β-induced *FN* and *COL1A1* production in DFs (Fig 5A and 5B). According to these findings, high NaCl environment decreased the collagen deposition of DFs, as well (Fig 5C).

## Discussion

The aim of the present study was to investigate the effect of excessive salt intake and the consequently increased dermal sodium content on skin pathology. Our results demonstrated for the first time that high salt loading influences inflammation related cytokine production, and the ECM remodeling of the skin.

The theory of salt storage in the skin was first mentioned more than 100 years ago but afterwards it was almost forgotten [28]. However, in the past two decades, largely due to the comprehensive investigations of Titze et al. our knowledge about the dermal sodium storage

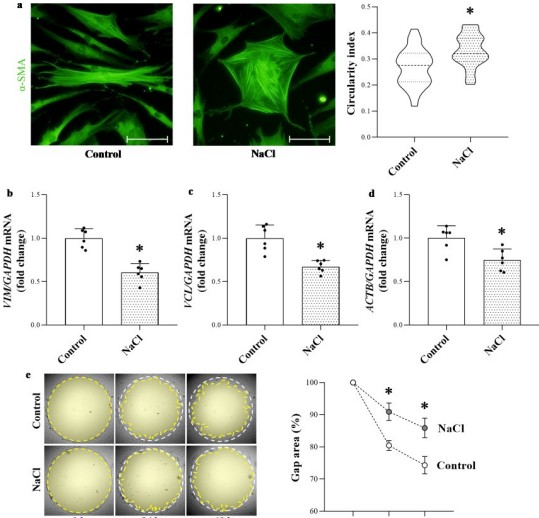

**Fig 4. Effect of high NaCl concentration on human primary dermal fibroblast cells.** (**a**) Cellular morphology of the cells was visualized by α-SMA immunofluorescent staining (green). Scale bar: 20 μm (**a**) Determination of circularity of α-SMA immunopositive human primary dermal fibroblasts was carried out by graphical analysis of the cells (n = 35–40). Values were expressed as mean+SD. *p<0.05 vs. Control (Unpaired t-test). (**b-d**) The mRNA expression of cell motility markers *VIM*, *VCL* and *ACTB* in the human dermal fibroblasts was determined by real-time PCR in comparison with *GAPDH*. Data were normalized and presented as the ratio of the mean values of the control group. Values were expressed as mean+SD. Dots represent individual values (n = 6). *p<0.05 vs. Control; (Unpaired t-test (b, c), Mann-Whitney U-test (d)). (**e**) The migration of DFs was determined by measuring the size of the cell-free gap area at 0, 24 and 48 hours (n = 8). Values were expressed as mean+SD. *p<0.05 vs. Control (two-way ANOVA + Sidak's multiple comparison).

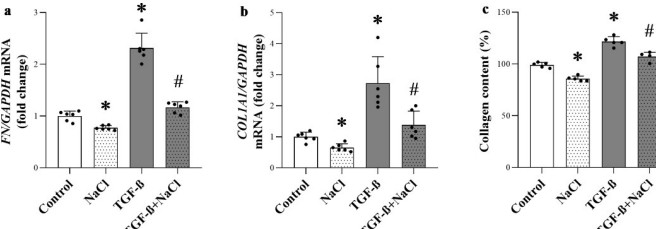

**Fig 5. Effect of high NaCl concentration on the ECM marker production of human primary dermal fibroblasts.** (**a-b**) The mRNA expression of *FN* and *COL1A1* in the human dermal fibroblasts was determined by real-time PCR in comparison with *GAPDH*. (**c**) The proportion of collagen content in human DFs was detected by SiriusRed collagen detection assay. Data were normalized and presented as the ratio of the mean values of the control group. Values were expressed as mean+SD (n = 6 in each group); *p<0.05 vs. Control; #p<0.05 vs. TGF-ß; (Unpaired t-test (b,c), Mann-Whitney U-test (a)).

became more comprehensive. Indeed, numerous experimental studies on rodents demonstrated that a significant part of the ingested dietary sodium is stored in the skin in osmotically active and inactive forms [2–9]. Moreover, sodium storage of the skin was also confirmed in humans by high resolution $^{23}$Na-MRI imaging [29]. Similarly, our present results also demonstrated that excess salt loading increases the dermal sodium content in mice (Fig 1C).

Recently, it has been demonstrated that local excess sodium facilitates the inflammatory cytokine production of immune cells and increases the severity of pre-existing pathological processes in various organs including kidney and gut, as well [4,12,14–16]. However, the role of increased dermal sodium content in skin pathologies has not been revealed yet. Therefore, in the present study we sought to investigate whether high salt diet resulting in increased dermal sodium content could influence the process of inflammation and the closely related tissue remodeling in healthy and IMQ-induced inflamed skin of mice *in vivo*.

Our results showed that high salt intake increased the expression of *Il17* in the skin of HSD mice (Fig 1E), which is in accordance with previous findings demonstrating that NaCl induces the activation of the immune system through its direct effects on the maturation of native T cells to form pro-inflammatory IL-17 producing Th17 cells [30].

Furthermore, IMQ treatment increased the dermal expression of many cytokines, including *Il1b*, *Il10*, *Il17* and *Tnfa*, however only the expression of *Il10* and *Il13* was altered by high salt diet of the IMQ treated mice (Fig 1D–1H).

IL-10 is one of the most important molecule in the anti-inflammatory cascade via its inhibitory effect on pro-inflammatory cytokine release and antigen presentation of various immune cells [31]. Similarly, IL-13 also has been demonstrated to reduce the activation of monocytes and macrophages [32,33]. Therefore, our results suggest that the HSD associated decreased expression of anti-inflammatory cytokine IL-10 and IL-13 production could lead to more severe dermal inflammation in mice treated with IMQ.

Based on the literary data, immune cells, including lymphocytes and monocytes are potential source of IL-10 and IL-13, therefore also we investigated whether high salt loading is responsible for the altered *IL10* and *IL13* expression of immune cells [34–36]. In accordance with our results in HSD and HSD IMQ animals, high salt loading reduced both *IL10* and *IL13* expression of PBMCs *in vitro* (Fig 2A and 2B). These results demonstrate the direct effect of high salt environment on the *IL10* and *IL13* production of immune cells and give possible explanation for their decreased expression in HSD animals.

Taken together, our results suggest that the high salt diet associated changes in expression of pro-inflammatory and anti-inflammatory cytokine expression may promote the inflammatory processes in the skin.

Inflammation plays a crucial role in dermal ECM remodeling since it is responsible for the induction of effector cells, growth factors and enzymes that orchestrate the balance between the production and degradation of ECM components during dermal remodeling. Therefore, we investigated whether the decreased presence of anti-inflammatory factors in HSD mice may be associated with altered expression of main cellular and structural elements of ECM. We found that expression of α-SMA *(Acta2)*, the specific marker of active DFs decreased in the skin of HSD IMQ mice compared to NSD IMQ mice (Fig 3A). Previously, it has been shown that dermal expression of α-SMA is proportional to the number and activity of DFs in the skin [37]. Furthermore, activated DFs are responsible for the production of ECM architecture components, including collagens and fibronectin [38]. Accordingly, we found decreased *Col1a1* and *Fn* expression in the skin of HSD IMQ mice compared to NSD IMQ mice (Fig 3B and 3C). These results suggest that under inflammatory conditions increased salt intake has an inhibitory effect on ECM production of DFs. Interestingly, although inflammation facilitates the production of ECM components in most organs, the expression of ECM decreased in the inflamed skin of mice on high salt diet.

Investigating the underlying molecular mechanisms of high salt intake related ECM alterations, we found decreased expression of *Pdgfb* in the skin of HSD and HSD IMQ mice compared to that of NSD controls (Fig 3D). PDGF-B plays a central role in the regulation of tissue remodeling as a strong mitogenic growth factor inducing both the proliferation and ECM production of DFs [39,40]. Therefore, it is easy to accept that decreased expression of α-SMA and ECM components in HSD IMQ mice is closely related to the decreased amount of the PDGF-B. Immune cells play a key role in growth factor production, including PDGF-B [41,42]. Accordingly, our *in vitro* experiments showed that high salt environment decrease the *PDGFB* production of immune cells, revealing a possible mechanism behind decreased *Pdgfb* production of HSD mice.

Our results demonstrated increased *Tgfb* expression in the IMQ treated groups which was not affected by HSD. TGF-ß has been considered as a main factor in the induction of myofibroblast differentiation and ECM component production. Interestingly, despite the increased *Tgfb* expression in HSD IMQ mice the *Acta2*, *Col1a1* and *FN* expression was significantly decreased compared to NSD IMQ mice (Fig 3A–3C and 3E). Since the TGF-ß induced SMAD transcription factors mediated signaling pathway is one of the main drivers of tissue remodeling process in almost all organ, we investigated the potential role of its central element, the activation of SMAD2/3 in the aforementioned controversy. Our results demonstrated that the amount of activated phosphorylated (p)-SMAD2/3 protein was significantly decreased in HSD IMQ group compared to NSD IMQ group (Fig 3F). Numerous previous studies demonstrated that reduced SMAD2/3 expression or pharmacological inhibition of SMAD signaling pathway decreases the expression of ECM remodeling markers [43]. Therefore, our results showing reduced pSMAD2/3 level in the skin of HSD IMQ mice may also explain why the expression of ECM remodeling genes was also reduced in these treatment group compared to that of NSD IMQ controls. Although TGF-ß has been regarded as the main driver of SMAD signaling, recent studies of ECM remodeling have revealed other cytokine mediated pathways that induce or facilitate SMAD2/3 activation [44,45]. Indeed, IL-13 was demonstrated to induce SMAD2/3 phosphorylation in hepatic stellate cells and enhance pSMAD2/3 activation in primary human fibroblasts as well [46,47]. In addition to the above, it is also important to note that the role of IL-13 in dermal tissue remodeling process has been widely demonstrated. Indeed, it has been shown that IL-13 is a potent inducer of activation and collagen synthesis of DFs [48–50]. Since our results also demonstrated decreased *Il13* expression in the HSD groups (Fig 1H), we hypothesize that the reduced *Il13* expression in HSD IMQ animals may contribute to reduced SMAD pathway activity, and decreased DF activation and ECM expression respectively.

Based on our *in vivo* results demonstrating the inhibitory effect of high salt loading on expression of ECM components and related factors we suggest that high salt intake may lead to impaired dermal repair. In accordance with this hypothesis, Binger et al. found delayed wound closure in mice on high salt diet. However, their results suggested that the decreased dermal M2 macrophage activity of mice on high salt diet is responsible for the delayed wound healing [20].

Although there are many different dermal cell types, the main source of different ECM components is almost exclusively the DFs in the skin, therefore our further experiments also investigated the direct effect of high salt loading on the functional activity of DFs *in vitro*. We experienced that high salt condition significantly influenced the morphology of DFs. Indeed, we found more circular shaped DFs in the high salt treated group (Fig 4A). The migratory status of the fibroblasts can be characterized by their shapes. While the circular DFs that spread on a flat surface are in stationary state, their elongated shape is associated with increased motility [51]. Therefore, the increased circularity of DFs in a high salt content milieu indicates their decreased motility. Accordingly, our results showed that high salt environment reduced the expression of cell motility markers, including *ACTB*, *VIM* and *VCL* in DFs (Fig 4B–4D). Beta-actin is a main cellular cytoskeletal component whose polymerization provides protrusive forces that push the cell forward [52]. More-over, vimentin and vinculin are proteins responsible for the cell adhesion to ECM components which supports the migration of DFs to the site of injury [53]. In line with the decreased expression of cell motility regulators, our *in vitro* migration assay demonstrated the decreased migration of DFs in high salt environment (Fig 4E). Since migration of human dermal fibroblasts is critical for wound contraction and repair, our results indicate that increased sodium concentration may impair the motility of DFs, which can disturb wound repair mechanisms. Furthermore, these results are in accordance with previous observations demonstrating reduced migratory capacity of myocardial and endothelial cells in high salt condition [54,55].

Furthermore, in line with our above experiment showing decreased *Fn* and *Col1a1* expression in HSD IMQ mice (Figs 3B and 4C), we found that high salt environment reduced the endogen *COL1A1* and *FN* expression, as well as the general ECM production of DFs *in vitro* (Fig 5A–5C). The activity of TGF-ß was also inhibited by high salt condition (Fig 5A–5C). These results suggest that increased local salt concentration could directly influence the dermal ECM production of DF cells. Our recent study demonstrated that high salt environment increases the prostaglandin E2 production of DFs, that has been shown to inhibit the DF's collagen expression in an autocrine TGF-β/Smad pathway-dependent manner, which gives a possible explanation for our present results [10,56].

In conclusion, our data suggest that excessive salt intake associated increased dermal sodium content may influence dermal inflammation and tissue remodeling. We described the pro-inflammatory effect of high salt loading, mediated by the alteration of local *Il17*, *Il13* and *Il10* expression *in vivo*. We demonstrated the inhibitory effect of high salt environment on the *IL10*, *IL13* and *PDGFB* production of immune cells. Our results also demonstrated that high salt consumption contributes to decreased dermal fibroblast activation and ECM production under inflammatory conditions and reduces the expression of ECM production related factor *Pdgfb* and decreases the activation of SMAD2/3 transcription factor. Finally, we also described the direct inhibitory role of high sodium environment on the dermal fibroblast activity *in vitro*. Our observations raise the possibility that excess salt content in diet is a potential risk factor for skin pathology associated with inflammation and impaired or chronic wound healing (Fig 6).

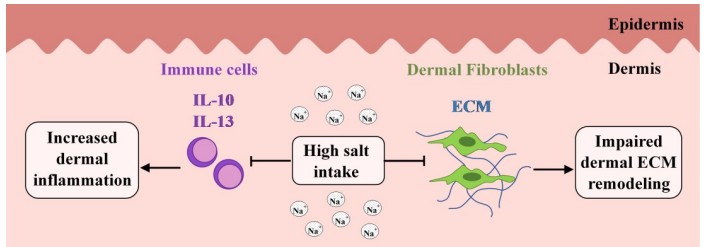

**Fig 6. Schematic representation of how high salt diet may mediate inflammation and ECM remodeling in the skin.** Our results showed that high salt loading and the consequently increased dermal sodium (Na⁺) content reduced the anti-inflammatory IL-10 and IL-13 cytokine expression of immune cells in the skin, even in dermatitis. These findings suggest the role of high salt diet in dermal inflammation. Furthermore, our results revealed the inhibitory effect of high salt loading on the ECM production of dermal fibroblast which can lead to impaired dermal ECM remodeling.

## Supporting information

**S1 Fig. Effect of 8% high salt (NaCl) diet on body weight.** Mice were kept on NSD or HSD for 14 days. The body weight was measured in every 7 days for two weeks. (n = 6 in each group).
(TIF)

**S1 Raw images. Images of entire Western blot membranes belong to Fig 3F.** Internal control (IC).
(PDF)

## Acknowledgments

We are grateful to Mária Bernáth for her excellent technical assistance.

## Author Contributions

**Data curation:** Csenge Pajtók.

**Investigation:** Csenge Pajtók, Apor Veres-Székely, Róbert Agócs, Beáta Szebeni.

**Methodology:** Apor Veres-Székely, Péter Dobosy, István Németh, Zoltán Veréb, Lajos Kemény.

**Resources:** Attila J. Szabó, Tivadar Tulassay.

**Supervision:** Domonkos Pap.

**Writing – review & editing:** Attila J. Szabó, Ádám Vannay, Tivadar Tulassay, Domonkos Pap.

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
