## [Decision Letter · Decision Letter 0]

23 Jul 2021

PONE-D-21-11775

High salt intake impairs dermal tissue remodeling

PLOS ONE

Dear Dr. Pap,

Thank you for submitting your manuscript to PLOS ONE. After careful consideration, we feel that it has merit but does not fully meet PLOS ONE’s publication criteria as it currently stands. Therefore, we invite you to submit a revised version of the manuscript that addresses the points raised during the review process.

1. High salt did not affect TGF-bata expression even in the IMQ treatment but down stream collagen expression was significantly decreased. Authors may study TGFbeta down stream molecules like smad2/3 expression which transcription factors can directly binding to the promotor region of the target genes like col1a1.

2. Figure legends: presented qPCR result description is not clear. "Values were expressed as mean + SD" but graph presented as a fold changes. Please edit text clearly described.

3. Statistical analysis: please clearly describe in the figure legend which experiment analyzed by t-test or U-test. What is the consideration for using each test in each experiment?

We look forward to receiving your revised manuscript.

Kind regards,

Minghua Wu, M.D., Ph.D.

Academic Editor

PLOS ONE

Journal Requirements:

2. Thank you for including your ethics statement: "All experiments on mice were approved by the Institutional Committee on Animal Welfare (PEI/001/1731-8/2015). Collection of biospecimens for the isolation and culture on human dermal fibroblasts complied with the directive of the Helsinki Declaration and were approved by the local Ethics Committee of the University of Szeged, Hungary (CSR/039/00346-5/2015). Isolation and culture on human PBMCs were approved by Semmelweis University Regional and Institutional Committee of Science and Research Ethics (31224‑5/2017/EKU)."

a) Please provide additional details regarding participant consent. In the ethics statement in the Methods and online submission information, please ensure that you have specified (1) whether consent was informed and (2) what type you obtained (for instance, written or verbal, and if verbal, how it was documented and witnessed). If your study included minors, state whether you obtained consent from parents or guardians. If the need for consent was waived by the ethics committee, please include this information.

3. Please consider modifying your title to ensure that it is specific, descriptive, and concise (for example by specifying the animal model used).

Reviewers' comments:

Reviewer's Responses to Questions

**Comments to the Author**

1. Is the manuscript technically sound, and do the data support the conclusions?

Reviewer #1: Partly

2. Has the statistical analysis been performed appropriately and rigorously? 

Reviewer #1: Yes

3. Have the authors made all data underlying the findings in their manuscript fully available?

Reviewer #1: Yes

4. Is the manuscript presented in an intelligible fashion and written in standard English?

Reviewer #1: Yes

5. Review Comments to the Author

Reviewer #1: Pajtók et al. examined the effects of high salt intake on skin remodeling in IMQ-treated mice. The manuscript is well written and includes novel findings, however, the authors should address some points as described below.

1. In this reviewer’s experience, when we used an 8% NaCl diet in mice, they don’t eat much maybe because of the taste, therefore we usually use a 4% NaCl diet as a high salt diet. Did the authors measure food intake and body weight in this study? 8% NaCl diet may reduce food intake, which can affect the skin remodeling independently of salt.

2. Based on the authors’ data, the main effect of high salt on cytokines in the IMQ model is IL10 and IL13, not IL17. Thus, figure 6 is confusing. Although the previous studies showed that the clear effect of salt on IL17, it is not clear in the IMQ model. This reviewer suggests that the authors should revise this point in the manuscript and figure 6.

6. PLOS authors have the option to publish the peer review history of their article (what does this mean?). If published, this will include your full peer review and any attached files.

Reviewer #1: No

---

## [Author Response · Author response to Decision Letter 0]

23 Sep 2021

Responses to Editor:

1. High salt did not affect TGF-beta expression even in the IMQ treatment but down stream collagen expression was significantly decreased. Authors may study TGFbeta down stream molecules like smad2/3 expression which transcription factors can directly binding to the promotor region of the target genes like col1a1.

Thank you for your suggestion. We agree that it is important to better understand the high salt intake induced molecular mechanisms which are responsible for the impaired dermal tissue remodeling. The TGF-ß induced and various SMAD transcription factors mediated signaling pathway is one of the main driver of tissue remodeling process in almost all organ. During its activation the phosphorylated SMAD2/3 molecules translocate to the cell nucleus of fibroblasts and promote gene expression changes that leads to cell differentiation into α-SMA positive myofibroblasts and to production of ECM elements including various collagens and fibronectin [1]. Interestingly, our results showed that while the expression of TGF-ß was increased in the skin of IMQ treated mice on high salt diet (HSD IMQ group) the expression of tissue remodeling alpha smooth muscle actin (ACTA2), fibronectin (FN) and collagen 1a1 (Col1a1) was decreased. 

In accordance with your suggestion we investigated the potential role of SMAD pathway activation behind this controversy. To this purpose we determined the amount of phosphorylated (p)SMAD2/3 in the skin samples using Western blot analysis. Our results demonstrated that the amount of pSMAD2/3 was significantly decreased in HSD IMQ group compared to NSD IMQ group (Fig 3/f).

Numerous previous studies demonstrated that reduced SMAD2/3 expression or pharmacological inhibition of SMAD signaling pathway decreases the expression of these ECM remodeling markers and tissue fibrosis [2]. Therefore, our results showing reduced pSMAD2/3 level in the skin of HSD IMQ mice may explain why the expression of ECM remodeling genes was also reduced in these treatment group compared to that of NSD IMQ controls.

Although TGF-ß has been regarded as the main drivers of SMAD signaling, recent studies of ECM remodeling has revealed other pathways that mediates SMAD2/3 activation [3, 4]. Indeed, IL-13 was demonstrated to induce SMAD2/3 phosphorylation in hepatic stellate cells and enhance pSMAD2/3 activation in primary human fibroblasts as well [5, 6]. In addition, IL-13 was demonstrated to facilitate tissue fibrosis in a completely TGF-ß independent manner [7]. Since our results showed decreased IL-13 expression in the skin of HSD IMQ animals compared to NSD IMQ animals, we hypothesize that the reduced IL-13 expression in HSD IMQ animals may contribute to reduced SMAD2/3 activity. Our new results are incorporated into the new version of the manuscript (page 17-18, line 384-408). Changes are highlighted in blue.

References:

1. Wynn TA. Cellular and molecular mechanisms of fibrosis. The Journal of Pathology. 2008;214(2):199-210. doi: https://doi.org/10.1002/path.2277.

2. Walton KL, Johnson KE, Harrison CA. Targeting TGF-β Mediated SMAD Signaling for the Prevention of Fibrosis. Frontiers in Pharmacology. 2017;8:461.

3. Derynck R, Zhang YE. Smad-dependent and Smad-independent pathways in TGF-β family signalling. Nature. 2003;425(6958):577-84. doi: 10.1038/nature02006.

4. Nguyen JK, Austin E, Huang A, Mamalis A, Jagdeo J. The IL-4/IL-13 axis in skin fibrosis and scarring: mechanistic concepts and therapeutic targets. Archives of Dermatological Research. 2020;312(2):81-92. doi: 10.1007/s00403-019-01972-3.

5. Liu Y, Meyer C, Müller A, Herweck F, Li Q, Müllenbach R, et al. IL-13 Induces Connective Tissue Growth Factor in Rat Hepatic Stellate Cells via TGF-β–Independent Smad Signaling. The Journal of Immunology. 2011;187(5):2814. doi: 10.4049/jimmunol.1003260.

6. Zhou X, Hu H, Huynh M-LN, Kotaru C, Balzar S, Trudeau JB, et al. Mechanisms of tissue inhibitor of metalloproteinase 1 augmentation by IL-13 on TGF-β1–stimulated primary human fibroblasts. Journal of Allergy and Clinical Immunology. 2007;119(6):1388-97. doi: https://doi.org/10.1016/j.jaci.2007.02.011.

7. Kaviratne M, Hesse M, Leusink M, Cheever AW, Davies SJ, McKerrow JH, et al. IL-13 Activates a Mechanism of Tissue Fibrosis That Is Completely TGF-β Independent. The Journal of Immunology. 2004;173(6):4020. doi: 10.4049/jimmunol.173.6.4020.

2. Figure legends: presented qPCR result description is not clear. "Values were expressed as mean + SD" but graph presented as a fold changes. Please edit text clearly described.

Thank you for your comment. In accordance with your suggestion, the texts of the figure legends were revised to be more clear. The mRNA expressions were determined by comparison with the expression of GAPDH or RPLP0 as a housekeeping gene. Data were normalized and presented as the ratio of the mean values of their control groups. Accordingly, we clarified the sections describing the qPCR results in the figure legends. Changes are highlighted in blue.

3. Statistical analysis: please clearly describe in the figure legend which experiment analyzed by t-test or U-test. What is the consideration for using each test in each experiment?

In statistical tests, we examined the effect of increased salt intake on gene expression changes under normal and pathological conditions in in vivo and in vitro model systems. In addition, we also examined the changes in gene expression induced by the models itself including in vivo IMQ treatment and in vitro TGF-ß treatment.

Therefore, we sought answers to the following three questions:

- Does salt loading cause significant changes in inflammatory or profibrotic gene expression in the skin in vivo (NSD↔ HSD) and in dermal fibroblasts and immune cells in vitro (Control↔ NaCl)?

- Does salt loading cause a significant change in inflammatory or profibrotic gene expression in IMQ-induced dermatitis in vivo (NSD IMQ↔ HSD IMQ) and in TGFβ-treated dermal fibroblasts in vitro (TGFβ↔ NaCl+TGFβ)?

- Do our in vivo (NSD↔ NSD IMQ) and in vitro (Control↔ TGFβ) models cause any inflammatory or profibrotic gene expression changes?

In each case, always two groups were compared. First, we determined whether the groups show a normal distribution by Kolmogorov-Smirnov test. Then, depending on whether the group had a normal or non-normal distribution, an unpaired t-test or Mann-Whitney U test was used to determine significance. Multiple comparisons of the raw data derived from migration assays were performed by using multiple t-test and ordinary two-way ANOVA with Dunnett correction. In all cases, p≤0.05 was considered as statistically significant. The figure legends were supplemented with a description of the exact statistical test used in that experiments. In addition, graphs were remade for better interpretation. Dots on the graphs represent individual values in order to show the distribution of the data.

Journal Requirements:

Both human dermal fibroblasts and human peripheral blood mononuclear cells were obtained from healthy individuals after signing an informed consent. We supplemented both the Methods and the ethics statement with this information. Changes are highlighted in blue.

3. Please consider modifying your title to ensure that it is specific, descriptive, and concise (for example by specifying the animal model used).

 Thank you for your comment. According to your suggestion, we modified the title of the manuscript to make it more specific. The new title is „High salt intake impairs dermal tissue remodeling in a mouse model of IMQ induced dermatitis”. Changes are highlighted in blue.

Responses to Reviewer:

1. In this reviewer’s experience, when we used an 8% NaCl diet in mice, they don’t eat much maybe because of the taste, therefore we usually use a 4% NaCl diet as a highsalt diet. Did the authors measure food intake and body weight in this study? 8% NaCl diet may reduce food intake, which can affect the skin remodeling independently of salt.

Thank you for your question. The high salt content of the chow can indeed affect the appetite of the animals and thus their body weight. In our preliminary experiments we investigated whether the high salt diet (HSD) containing 8% NaCl has an effect on the body weight of mice. Our results showed that there was no significant difference in body weight between animals on normal salt diet (NSD) and those on HSD even after two weeks, therefore we terminated the experiment. The graphical representation of this experiment was included in Supporting Information (S2 Fig).

Although the duration of our preliminary experiment was shorter than the study design we used in the manuscript (14 days vs. 33 days), our data are in line with recent studies showing no difference in the body weight of mice and rats on HSD containing 8% NaCl even after 10 weeks compared to control animals on NSD [1-4]. These results are incorporated into the new version of the manuscript (page 10, line 220-222). Changes are highlighted in blue.

It is also important to note that our results showed that HSD alone had no effect on skin ECM remodeling. Indeed, there was no difference in the expression of ECM remodeling markers, including alpha smooth muscle actin (ACTA2), fibronectin (FN) and collagen1a1 (Col1a1) between mice kept on NSD and HSD alone. However, under inflammatory conditions induced by the imiquimod (IMQ) treatment the high salt intake resulted in decreased ECM production in HSD IMQ group compared to NSD IMQ group (Fig. 3/a-c). Therefore, our results suggest that high salt consumption exerts inhibitory effect on skin ECM remodeling under inflammatory conditions.

References:

1. Faraco G, Brea D, Garcia-Bonilla L, Wang G, Racchumi G, Chang H, et al. Dietary salt promotes neurovascular and cognitive dysfunction through a gut-initiated TH17 response. Nature Neuroscience. 2018;21(2):240-9. doi: 10.1038/s41593-017-0059-z.

2. Ferguson JF, Aden LA, Barbaro NR, Van Beusecum JP, Xiao L, Simmons AJ, et al. High dietary salt–induced DC activation underlies microbial dysbiosis-associated hypertension. JCI Insight. 2019;4(13). doi: 10.1172/jci.insight.126241.

3. Guo C-P, Wei Z, Huang F, Qin M, Li X, Wang Y-M, et al. High salt induced hypertension leads to cognitive defect. Oncotarget. 2017;8(56):95780-90. doi: 10.18632/oncotarget.21326. PubMed PMID: 29221166.

4. Ogihara T, Asano T, Ando K, Chiba Y, Sekine N, Sakoda H, et al. Insulin Resistance With Enhanced Insulin Signaling in High-Salt Diet–Fed Rats. Diabetes. 2001;50(3):573. doi: 10.2337/diabetes.50.3.573.

2. Based on the authors’ data, the main effect of high salt on cytokines in the IMQ model is IL10 and IL13, not IL17. Thus, figure 6 is confusing. Although the previous studies showed that the clear effect of salt on IL17, it is not clear in the IMQ model. This reviewer suggests that the authors should revise this point in the manuscript and figure 6.

Thank you for your remark. Although the inducing effect of increased salt intake on dermal proinflammatory IL-17 cytokine expression is one of our novel findings, the high salt diet (HSD) indeed did not affect its expression in mice receiving IMQ treatment. Therefore, in accordance with your suggestion we removed the IL-17 related parts of Figure 6. In accordance with this, we also removed the parts of Figure 6 showing the inhibitory effect of high salt environment on motility of dermal fibroblast since these in vitro results were not investigated in our in vivo model. We also revised the figure legend of Figure 6. Please find the revised caption of Figure 6 on page 20, line 457-461.

---

## [Decision Letter · Decision Letter 1]

29 Sep 2021

High salt diet impairs dermal tissue remodeling in a mouse model of IMQ induced dermatitis

PONE-D-21-11775R1

Dear Dr. Pap,

We’re pleased to inform you that your manuscript has been judged scientifically suitable for publication and will be formally accepted for publication once it meets all outstanding technical requirements.

Kind regards,

Minghua Wu, M.D., Ph.D.

Academic Editor

PLOS ONE

Additional Editor Comments (optional):

Reviewers' comments:

Reviewer's Responses to Questions

**Comments to the Author**

1. If the authors have adequately addressed your comments raised in a previous round of review and you feel that this manuscript is now acceptable for publication, you may indicate that here to bypass the “Comments to the Author” section, enter your conflict of interest statement in the “Confidential to Editor” section, and submit your "Accept" recommendation.

Reviewer #1: All comments have been addressed

2. Is the manuscript technically sound, and do the data support the conclusions?

Reviewer #1: Yes

3. Has the statistical analysis been performed appropriately and rigorously? 

Reviewer #1: Yes

4. Have the authors made all data underlying the findings in their manuscript fully available?

Reviewer #1: Yes

5. Is the manuscript presented in an intelligible fashion and written in standard English?

Reviewer #1: Yes

6. Review Comments to the Author

Reviewer #1: (No Response)

7. PLOS authors have the option to publish the peer review history of their article (what does this mean?). If published, this will include your full peer review and any attached files.

Reviewer #1: No

---

## [Editor Report · Acceptance letter]

15 Oct 2021

PONE-D-21-11775R1 

High salt diet impairs dermal tissue remodeling in a mouse model of IMQ induced dermatitis 

Dear Dr. Pap:

I'm pleased to inform you that your manuscript has been deemed suitable for publication in PLOS ONE. Congratulations! Your manuscript is now with our production department. 

Kind regards, 

on behalf of

Dr. Minghua Wu 

Academic Editor

PLOS ONE